# Emerging Molecular and Synaptic Targets for the Management of Chronic Pain Caused by Systemic Lupus Erythematosus

**DOI:** 10.3390/ijms25073602

**Published:** 2024-03-22

**Authors:** Han-Rong Weng

**Affiliations:** Department of Basic Sciences, California Northstate University College of Medicine, Elk Grove, CA 95757, USA; han-rong.weng@cnsu.edu; Tel.: +1-916-686-7897

**Keywords:** rheumatological, nociception, HCA2, DRG, cytokines, neuroinflammation

## Abstract

Patients with systemic lupus erythematosus (SLE) frequently experience chronic pain due to the limited effectiveness and safety profiles of current analgesics. Understanding the molecular and synaptic mechanisms underlying abnormal neuronal activation along the pain signaling pathway is essential for developing new analgesics to address SLE-induced chronic pain. Recent studies, including those conducted by our team and others using the SLE animal model (*MRL*/*lpr* lupus-prone mice), have unveiled heightened excitability in nociceptive primary sensory neurons within the dorsal root ganglia and increased glutamatergic synaptic activity in spinal dorsal horn neurons, contributing to the development of chronic pain in mice with SLE. Nociceptive primary sensory neurons in lupus animals exhibit elevated resting membrane potentials, and reduced thresholds and rheobases of action potentials. These changes coincide with the elevated production of TNFα and IL-1β, as well as increased ERK activity in the dorsal root ganglion, coupled with decreased AMPK activity in the same region. Dysregulated AMPK activity is linked to heightened excitability in nociceptive sensory neurons in lupus animals. Additionally, the increased glutamatergic synaptic activity in the spinal dorsal horn in lupus mice with chronic pain is characterized by enhanced presynaptic glutamate release and postsynaptic AMPA receptor activation, alongside the reduced activity of glial glutamate transporters. These alterations are caused by the elevated activities of IL-1β, IL-18, CSF-1, and thrombin, and reduced AMPK activities in the dorsal horn. Furthermore, the pharmacological activation of spinal GPR109A receptors in microglia in lupus mice suppresses chronic pain by inhibiting p38 MAPK activity and the production of both IL-1β and IL-18, as well as reducing glutamatergic synaptic activity in the spinal dorsal horn. These findings collectively unveil crucial signaling molecular and synaptic targets for modulating abnormal neuronal activation in both the periphery and spinal dorsal horn, offering insights into the development of analgesics for managing SLE-induced chronic pain.

## 1. Introduction

Systemic lupus erythematosus (SLE) is a chronic autoimmune disease marked by the dysregulation of adaptive and innate immunity. Consequently, the loss of self-tolerance with the formation of nuclear autoantigens and immune complexes results in inflammation and damage to multiple organs [1,2,3]. Women of childbearing age are at the greatest risk of developing SLE although it can affect people of all ages and males as well. It is estimated that there are 43.7 adults with SLE per 100,000 persons, corresponding to approximately 3.41 million adults worldwide [4]. Despite significant advancements in medical management that have notably reduced mortality rates associated with SLE, chronic pain persists in as many as 90% of SLE patients [5,6,7]. Pain is one of the most common symptoms in patients with SLE [8,9]. Pain experienced by patients with SLE can stem from various sources, including arthritis [10,11], tendonitis [11,12], nephritis [13], peripheral neuropathy [14,15,16], vasculitis [17], myopathy [18], serositis such as peritonitis [19], painful cutaneous manifestations [20], fibromyalgia [21,22], and SLE-related headaches [23]. While pain is a significant symptom during the active stage of SLE, recent studies showed that severe pain can also occur in SLE patients with mild and moderate SLE disease activities [24,25]. The current painkillers used for pharmacotherapeutic pain management, primarily non-steroidal anti-inflammatory drugs (NSAIDs) and opioids, are constrained by their efficacy and safety profiles. Consequently, numerous patients with SLE still endure chronic pain, greatly diminishing their quality of life [5,6,7]. The development of novel analgesics with high efficacy and safety profiles is an unmet need.

Understanding the cellular and molecular mechanisms underlying the genesis of chronic pain induced by SLE is a pivotal step towards the development of analgesics. Currently, animal model studies on chronic pain caused by SLE remain limited. Recent studies from our laboratory and others have revealed several signaling molecules implicated in altered pain signaling in the periphery and spinal dorsal horn of SLE mouse models [26,27,28,29]. In this review, I have summarized recent advancements in our comprehension of the molecular and synaptic mechanisms driving peripheral and spinal central sensitization in a mouse model of SLE with chronic pain. It is anticipated that these studies lay the groundwork and offer pathways for the development of innovative analgesics to manage chronic pain induced by SLE.

## 2. Animal Models Used for the Study of Chronic Pain Induced by SLE and Their Nociceptive Behavioral Phenotype

Murine models serve as the cornerstone of experimental animals utilized to identify potential therapeutic targets within the various signaling pathways dysregulated in SLE [30,31,32]. A range of mouse models for SLE, encompassing those that spontaneously develop the condition, as well as those for whom the condition was induced or genetically engineered, have been extensively employed in research endeavors [30,31]. In recent years, we and others have endeavored to pinpoint molecular targets for managing chronic pain resulting from SLE. To explore the molecular mechanisms underpinning SLE-induced chronic pain, we utilized *MRL* lupus-prone (*MRL*/*lpr*) mice—an established mouse model for SLE—as well as *MRL* normal control (*MRL*/*MpJ*) mice [26,27,28,29]. The *MRL*/*lpr* lupus-prone mouse serves as a widely recognized mouse model for human SLE [33,34,35]. *MRL*/*lpr* mice, which carry a mutation in the apoptosis-related Fas gene, spontaneously develop a severe autoimmune disease that mimics many features as in SLE patients, including immune abnormalities affecting T and B cells, autoantibody production, immune complex formation, and systemic inflammation [33,34]. As manifested in SLE patients, *MRL*/*lpr* mice have inflammation in multiple tissues (organs), including the joints [24,36,37,38], muscles [39], vascularies [40], kidneys [41,42], brain [43,44], and peripheral nerve [45]. These widespread sites of inflammation in the *MRL*/*lpr* mice are all potential sources of the chronic pain that develops in *MRL*/*lpr* mice. It was found that both male and female *MRL*/*lpr* mice spontaneously developed hypersensitivity in their hind paw to radiant heat stimuli from the age of 12–13 weeks, which reached a plateau between the age of 14 and 16 weeks old [26,27,28,29]. Changes in mechanical sensitivity in female MRL/lpr mice were evaluated using von Frey monofilaments, which showed that mechanical allodynia began to appear at the age of 14 weeks and persisted through the age of 16 weeks [27,29]. The development of ongoing pain and impaired wellbeing in female MRL/lpr mice were assessed by analyzing burrowing behaviors in the mice [29]. Burrowing is an innate behavior in rodents, which has been used to measure wellbeing and is also considered an index of ongoing pain in rodents [46,47,48,49,50]. It was observed that lupus mice at 14 weeks of age exhibited reduced burrowing activities compared to their activity levels at 10 weeks old, as well as compared to 14-week-old normal control mice. This suggests that *MRL*/*lpr* mice at 14 weeks old experience spontaneous ongoing pain and diminished wellbeing [29].

## 3. Nociceptive Primary Sensory Neurons in the Dorsal Root Ganglion from Lupus Mice with Chronic Pain Exhibit Hyperexcitability

Pathological pain originates from abnormal neuronal activation along the pain signaling pathway, which includes primary sensory neurons, neurons in the spinal dorsal horn, and supra-spinal areas. We recently reported that altered functionality in the peripheral primary sensory neurons makes an important contribution to the genesis of chronic pain caused by SLE. Sensory neurons located in the dorsal root ganglion (DRG) are responsible for transmitting peripheral sensory stimuli to the central nervous system. Primary sensory neurons are the first-order sensory neurons in the sensory transmission. They are pseudounipolar neurons, in which a single process emerging from their cell body bifurcates into a peripheral and a central branch [51]. Peripheral processes from sensory neurons innervate the skin, muscles, joints, and visceral organs [52]. Based on their responses to noxious stimuli, primary sensory neurons are divided into nociceptive neurons and non-nociceptive neurons. Nociceptive neurons are small neurons with two different types: one gives rise to thinly myelinated (Aδ) fibers, while another gives rise to unmyelinated (C) fibers [53]. Under normal conditions, upon peripheral nociceptive stimulation, ion channels in nociceptive receptors (nociceptors) at the nerve terminals from primary nociceptive neurons are open, resulting in an influx of cation ions into the neurons and depolarization of the membrane potential, comprising a process called nociceptive sensory transduction [54]. Once the membrane depolarization reaches the activation threshold of voltage-gated sodium channels, it triggers an influx of sodium ions through the voltage-gated sodium channels and generates an action potential. Action potentials then propagate along the sensory afferent fibers and transmit the nociceptive signal to the CNS [51]. Sensory neurons in the DRG have been recognized as the first processing station regulating nociceptive sensory transmission. Given that the generation and propagation of action potentials in sensory neurons are regulated by neuronal membrane electrophysiological properties, we quantitatively analyzed and compared membrane properties in nociceptive sensory neurons in the DRG of *MRL/MpJ* mice (normal control) mice and *MRL/Lpr* (lupus) mice with chronic pain [28]. 

### 3.1. Resting Membrane Potential Is Elevated while Capacitance Is Reduced in Nociceptive Primary Sensory Neurons of Lupus Mice with Chronic Pain

Using ex vivo whole-cell patch recording techniques, we compared excitability in nociceptive primary sensory neurons in the DRG from L4-L5 segments between male lupus mice with chronic pain and normal control mice [28]. It was found that neither lupus mice nor normal control mice had spontaneous action potentials in their primary nociceptive sensory neurons [28]. Passive membrane properties (i.e., resting membrane potential, membrane input resistance and capacitance) were analyzed. A normal resting membrane potential is essential to maintain a baseline electrical state and the overall excitability of a neuron, which is crucial to ensure that neurons to respond to stimuli. We found that resting membrane potentials in nociceptive primary sensory neurons from lupus mice with chronic pain were significantly elevated compared to those from control mice [28]. Membrane input resistance is a measure of number of open ion channels (or ion channel conductance), while membrane capacitance reflects the size of the cell and thickness of the cell membrane and affects the propagation velocity of action potentials [55]. We found that input resistances in neurons from lupus mice were similar to those in control mice, while membrane capacitances in lupus mice were significantly smaller than those in control mice [28].

### 3.2. Rheobase and Action Potential Threshold Are Reduced in Nociceptive Primary Sensory Neurons of Lupus Mice with Chronic Pain

Nociceptive primary sensory neurons from lupus mice with chronic pain exhibit different active membrane properties in comparison with those from normal controls [28]. The minimal currents required to evoke action potentials (rheobase) in the sensory neurons from lupus mice were significantly lower than those in neurons from control mice. Notably, the action potential thresholds of neurons from lupus mice were significantly lower than those in neurons from the control group. However, no significant differences were observed in the action potential amplitudes or half-action potential durations between control and lupus groups. Furthermore, the peak inward currents evoked by membrane depolarization between control and lupus groups were similar [28]. We also explored potential disparities in the action potential firing patterns in the nociceptive sensory neurons between lupus mice experiencing chronic pain and their control counterparts. In response to a depolarizing current pulse set at two times the rheobase current intensity (1 s duration), either a single action potential at the onset of the depolarizing current or multiple action potentials throughout the duration of the current injection were evoked in both control mice and lupus mice. The percentage of neurons displaying multiple action potentials in the lupus group did not show a significant difference compared to the control group [28]. Furthermore, it was found that the voltage-gated A-type K^+^ currents in lupus mice and normal control mice were similar, suggesting that voltage-gated A-type potassium channels are not involved in the reduction in action potential thresholds in lupus mice [28]. These findings indicate that the membrane properties of nociceptive sensory neurons in lupus mice exhibit both similarities and distinctive features when compared to those in other chronic pain animal models.

### 3.3. Comparison of Membrane Properties in Nociceptive Primary Sensory Neurons in Animals with Lupus-Induced Pain and Pain Induced by Other Diseases

The elevated resting membrane potentials observed in lupus mice resemble those found in animals experiencing neuropathic pain due to nerve ligation [56,57], spinal cord injury [58], or diabetes [59], and pain induced by arthritis [60] or bone cancer [61,62]. However, this trend is not observed in animals with inflammatory pain caused by complete Freund’s adjuvant (CFA) [63] (Table 1). Furthermore, nociceptive sensory neurons in lupus mice exhibit smaller capacitances compared to normal controls, contrary to previous findings in animals with CFA-induced inflammatory pain [63] or arthritis [64], where capacitances remained unchanged [63]. The reduction in action potential thresholds and rheobases in the nociceptive sensory neurons of lupus mice aligns with observations in animals with chronic pain induced by nerve injury [56,65], inflammation [60,63], or bone cancer [61,62]. Conversely, alterations in action potential durations, amplitudes, and firing patterns have been observed in animals experiencing nerve injury [56,65,66], inflammatory pain induced by CFA [63,67], and bone cancer pain [61,62]. In contrast, these parameters remain unaffected in lupus mice. Additionally, abnormal spontaneous action potential activity has been identified in nociceptive sensory neurons in animals with chronic pain from nerve injury [65], bone cancer pain [61], pain induced by arthritis [60,64], and inflammation [63]. However, no spontaneous firing was observed in the nociceptive sensory neurons of lupus mice. These findings suggest a distinct mechanism underlying peripheral sensitization in lupus disease. Nonetheless, the combination of elevated resting membrane potentials, a smaller membrane capacitance, and the reduced thresholds and rheobases of action potentials in nociceptive primary sensory neurons (Figure 1) in lupus mice collectively amplifies their responsiveness to low-threshold stimuli (i.e., peripheral sensitization) in vivo, ultimately leading to heightened pain perception.

## 4. Signaling Molecules Regulating the Excitability of Nociceptive Primary Sensory Neurons and Chronic Pain in Lupus Mice

Our recent studies have explored signaling molecules implicated in the regulation of excitability in nociceptive primary sensory neurons and chronic pain induced by SLE. These include sodium channel 1.7 (Nav1.7), tumor necrosis factor alpha (TNFα) and interleukin-1beta (IL-1β), extracellular signal-regulated kinase (ERK), and AMP-activated protein kinase (AMPK).

### 4.1. Protein Expression of Nav1.7 Is Not Altered in Nociceptive Primary Sensory Neurons in Lupus Mice with Chronic Pain

In order to understand the ion channel mechanisms underpinning the hyperexcitability in nociceptive primary sensory neurons in lupus mice, we measured the protein expression of Nav1.7 in sensory neurons in the DRG [28]. Nav1.7 is a primary voltage-gated sodium channel in nociceptive sensory neurons in the DRG, which is a key determinant of the action potential amplitude and threshold [68]. Nav1.7 is expressed in nociceptive primary sensory neurons, as well as at nerve fiber endings in the periphery and presynaptic terminals in the dorsal horn of the spinal cord [69]. We found that Nav1.7 protein expression in the L4–L5 spinal segments of the DRGs were similar between lupus mice with chronic pain and normal control mice. This finding is consistent with our patch clamp findings, which showed peak inward currents and action potential amplitudes in lupus mice comparable to those observed in control mice. The unchanged action potential amplitude and Nav1.7 protein expression in lupus mice, combined with a significant reduction in action potential thresholds in nociceptive sensory neurons, strongly suggest that the heightened excitability in nociceptive sensory neurons associated with chronic pain induced by SLE is likely due to alterations in voltage-gated mechanisms rather than changes in the protein expression levels of Nav1.7. In contrast, previous studies showed that animals with neuropathic pain [56], inflammatory pain [63], or surgical incision [70] exhibit increased action potential amplitude and Nav1.7 protein expression. 

### 4.2. Increased Production of Proinflammatory Mediators in the DRG Is Associated with Hyperexcitable Nociceptive Primary Sensory Neurons of Lupus Mice with Chronic Pain

It was reported that the protein expression of pro-inflammatory cytokines like TNFα and IL-1β, and ERK activity, were increased in the DRG in lupus mice with chronic pain, which were also associated with increased excitability in nociceptive sensory neurons in the DRG [28]. TNFα and IL-1β are prototypic proinflammatory cytokines. TNFα and IL-1β are known to be released from satellite glial cells [71], macrophages, and monocytes in the DRG of animals with nerve injury [72]. The heightened protein expression of TNFα and IL-1β in the DRG has been linked to increased excitability in nociceptive sensory neurons in animals experiencing chronic pain due to nerve injury [73,74,75], bone cancer [76], and arthritis [77,78]. Elevated ERK activity has been recognized as a marker for neuronal hyperexcitability [79], and a prominent event leading to nociceptor sensitization [80]. ERK was activated by TNFα in the DRG, mediating the facilitatory effects of cytokines on voltage-gated sodium channels [81]. In line with these studies, the heightened ERK activity, as well as the overproduction of TNFα and IL-1β in the DRG of lupus mice with chronic pain, further support the notion that neuroinflammation at the DRG may contribute to the genesis of hyperexcitability in the nociceptive sensory neurons and chronic pain in lupus mice.

### 4.3. AMPK Regulates Thermal Hyperalgesia in Lupus Mice

Given that AMPK is a known negative regulator for ERK activity and pro-inflammatory cytokine production in DRG neurons [82,83,84] and other cell types [85], we investigated the role of AMPK in regulating nociceptive sensory neuronal excitability and chronic pain in lupus mice [28]. AMPK is a crucial kinase involved in regulating cellular ATP homeostasis [86]. An increase in the intracellular AMP/ATP ratio following ATP consumption activates AMPK. AMPK activation leads to a reduction in ATP consumption, an increase in ATP production, and restoration of the AMP/ATP ratio [86]. Changes in AMPK activity in DRG sensory neurons and the spinal dorsal horn are implicated in the genesis of neuropathic pain [87,88,89], postoperative pain [90], and inflammatory pain induced by CFA [91] in rodents. It was found that lupus mice with chronic pain exhibited the significant suppression of phosphorylated AMPK protein expression in the DRGs compared to control mice, suggesting that the suppression of AMPK activity in the DRG may contribute to the genesis of chronic pain and hyperexcitability in nociceptive primary sensory neurons in lupus mice [28]. This notion was supported by both behavioral and electrophysiological experiments. Normal control mice receiving an intraplanar injection of a selective AMPK inhibitor (Compound C) [82] developed thermal hypersensitivity, as evidenced by a shortened latency of hind paw withdrawal responses to radiant heat stimuli 15–30 min after injection. Conversely, thermal hyperalgesia in lupus mice was ameliorated when a widely used AMPK activator, 5-amino-4-imidazole carboxamide (AICAR) [83], was subcutaneously injected into the plantar side of the hind paw [28]. These results underscored the role of AMPK in modulating thermal sensitivity and implicated its suppression in the manifestation of thermal hyperalgesia in mice with lupus disease.

### 4.4. AMPK Regulates the Hyperexcitability in Nociceptive Primary Sensory Neurons in Lupus Mice

The role of AMPK in regulating the excitability of nociceptive primary sensory neurons was directly demonstrated through ex vivo patch clamp recordings conducted on the DRG from normal control mice [28]. These recordings revealed that the pharmacological inhibition of AMPK activity, achieved by perfusing the AMPK inhibitor (Compound C) into the recording chamber, resulted in significant changes in the excitability of nociceptive primary sensory neurons. Specifically, there was a notable elevation in the resting membrane potential and a marked decrease in membrane capacitance, without affecting their input resistance. Moreover, the suppression of AMPK activity led to a significant reduction in the rheobase and the activating threshold for action potentials [28]. However, it is worth noting that Compound C did not induce significant alterations in the amplitude and duration of action potentials in the nociceptive primary sensory neurons. Additionally, the inhibition of AMPK activity in nociceptive DRG neurons did not produce significant changes in spontaneous action potential activities or in their firing patterns [28]. These findings collectively suggest that the suppression of AMPK in normal nociceptive sensory neurons mirrors the membrane electrical properties observed in nociceptive sensory neurons in lupus mice experiencing chronic pain. Thus, it is conceivable that the diminished AMPK activity in nociceptive primary sensory neurons enhances their excitability, contributing to the development of chronic pain in lupus mice.

#### 4.4.1. Molecular Mechanisms Underlying the Regulation of AMPK on the Resting Membrane Potentials in Nociceptive Sensory Neurons in Lupus Mice with Chronic Pain

Resting membrane potentials in DRG nociceptive neurons are influenced by the conductance and activities of several channels, including K2P “leak” channels, M channels (Kv7, KCNQ), 4-AP-sensitive KV channels (such as Kv1.4, Kv2s, and Kv3.4), and the Na^+^-K^+^ ATPase pump [92]. A reduction in the conductance of these channels would increase membrane resistance and decrease outward cationic current, thereby elevating the resting membrane potential.

Previous studies have shown that animals experiencing neuropathic pain exhibited decreased functioning of TREK2, a significant subtype of K2P channels, within the cell membrane of nociceptive sensory neurons. This reduction in function led to decreased conductance in K2P “leak” channels and the subsequent elevation of resting membrane potentials [93,94]. However, in the case of DRG nociceptive neurons in lupus mice, where an elevation of resting membrane potentials was observed without significant changes in membrane resistance, it was improbable that the elevated potentials were caused by the reduced conductance to the outward cationic current in the membrane channels. Instead, it was suggested that a reduction in the function of the Na^+^-K^+^ ATPase pump induced by low AMPK activity might be attributed to the elevated resting membrane potential in lupus mice [28] (Figure 2). Several lines of previous research support this notion: a. the suppression of Na^+^-K^+^ ATPase activity has been shown to increase neuronal excitability [95], while enhanced Na^+^-K^+^ ATPase activity resulted in the hyperpolarization of resting membrane potential without changes in membrane resistance (conductance) [96]; b. studies conducted on various cell lines have demonstrated that AMPK served as an activator of Na^+^-K^+^ ATPase [97,98]; c. lupus mice exhibited low AMPK activity in the DRG, and the inhibition of AMPK led to the depolarization of the resting membrane potential in nociceptive sensory neurons in the DRGs, without significant changes in membrane resistance.

#### 4.4.2. Molecular Mechanisms Underlying the Regulation of Action Potential Thresholds by AMPK in Nociceptive Sensory Neurons in Lupus Mice with Chronic Pain

The activation thresholds of action potentials in nociceptive sensory neurons are intricately regulated by various ion channels, including voltage-gated A-type K^+^ channels and voltage-gated Na^+^ channels. Voltage-gated A-type K^+^ channels function as a regulatory mechanism to counteract membrane depolarization, thereby modulating the action potential activation threshold and its waveform in sensory neurons [99]. Interestingly, the similarity observed in voltage-gated A-type K^+^ currents between lupus mice with chronic pain and control mice suggested that these channels may not be involved in the reduction in action potential thresholds observed in lupus mice. Among the channels implicated in action potential threshold regulation, Nav1.7 channels play a predominant role in the activation of DRG nociceptive neurons [68]. Previous studies have demonstrated that the phosphorylation of Nav1.7 channels by extracellular signal-regulated kinase (ERK) leads to a decrease in the activation thresholds of voltage-gated Nav1.7 channels [100]. ERK serves as a pivotal signaling transducer activated by numerous pro-inflammatory cytokines, including IL-1β and TNFα, and has been implicated in hyperexcitability in nociceptive primary sensory neurons, induced by nerve injury and inflammation [101,102]. Consistent with previous findings, the abundance of phosphorylated ERK, IL-1β, and TNFα proteins in the DRG of lupus mice was notably higher compared to normal controls. Moreover, we have shown that AMPK is a downstream signaling molecule used by IL-1β in the spinal cord [89]. The activation of AMPK suppresses ERK activation [88,103], whereas the inhibition of AMPK with Compound C leads to ERK phosphorylation [100]. Therefore, it is conceivable that the suppression of AMPK activity induced by the enhanced IL-1β, and TNFα in sensory neurons in lupus mice may result in ERK activation, subsequently leading to the phosphorylation of sodium channel Nav1.7, and thereby lowering the action potential activation threshold in nociceptive sensory neurons [28] (Figure 2).

## 5. Lupus Mice with Chronic Pain Have Enhanced Glutamatergic Synaptic Transmission at the First Sensory Synapse in the Spinal Dorsal Horn

The spinal dorsal horn serves as a crucial relay center for sensory transmission, where it receives and processes sensory inputs originating from the periphery. Noxious sensory inputs from the periphery travel along primary sensory fibers and enter the spinal dorsal horn. Within this region, excitatory neurotransmitters, primarily glutamate, are released from the central terminals of primary afferents. The binding of glutamate to ionotropic glutamate receptors on postsynaptic neurons in the spinal dorsal horn triggers the opening of glutamate-ligand-gated channels, leading to the influx of Na^+^ and Ca^2+^ ions into the postsynaptic neurons and the subsequent depolarization of membrane potentials. Upon reaching the activating threshold of voltage-gated Na^+^ channels, action potentials are generated in postsynaptic neurons, thereby completing the signal transmission at the first synapse along the pain sensory pathway. Recent investigations conducted by our team and others have unveiled alterations in spinal glutamatergic synaptic transmission in female lupus mice with chronic pain. Additionally, these studies have identified signaling molecules that regulate plasticity at the spinal glutamatergic synapses in female lupus mice.

The activation of glutamatergic synapses plays a pivotal role in neuronal activation within the central nervous system, including the spinal cord. This activation is governed by three key factors: the quantity of glutamate release from presynaptic terminals, the functionality of glutamatergic receptors at postsynaptic neurons, and the rate of glutamate clearance at the synaptic cleft. Since there are no extracellular enzymes capable of metabolizing released glutamate, the clearance and maintenance of glutamate homeostasis rely on glutamate transporters, which are expressed in both neurons and astrocytes [104]. Our research has demonstrated that astrocytic glutamate transporters critically regulate the activation of α-Amino-3-hydroxy-5-methyl-4-isoxazolepropionic Acid (AMPA) and N-methyl-D-aspartate (NMDA) receptors at the first synaptic transmission along the pain signaling pathway in the spinal dorsal horn [105,106,107]. Utilizing whole-cell patch-clamp recording techniques, our team and others have recently revealed that the three key factors governing the activation of glutamatergic synapses were significantly altered in the spinal dorsal horn in female lupus mice experiencing chronic pain. Lupus mice with chronic pain exhibit heightened glutamate release from nociceptive primary afferents, as evidenced by the increased frequencies of miniature AMPA currents recorded from neurons in the spinal dorsal horn, which are monosynaptically connected to the primary nociceptive afferents. Additionally, the amplitudes of miniature AMPA currents in lupus mice were found to be elevated in comparison to those in normal control mice [29], indicating the enhanced functionality of postsynaptic AMPA receptors. Furthermore, we demonstrated impairments in glutamate uptake by astrocytes in the spinal dorsal horn of lupus mice, with a reduced protein expression of astrocytic glutamate transporters (GLT-1) [27]. The function of glial glutamate transporters was further analyzed in real time by recording glutamate transporter currents from astrocytes in the spinal superficial laminae [27,108]. Glutamate uptake by glial glutamate transporters involves the co-transport of two or three Na^+^ ions with one H^+^ ion and the counter-transport of one K^+^ ion, resulting in the translocation of a net positive charge during each transport cycle and the generation of currents termed glutamate transporter currents [109,110]. It was observed that these glutamate transporter currents in astrocytes situated in the spinal superficial laminae were notably diminished in lupus mice with chronic pain compared to normal control mice, suggesting a suppression of glutamate transporter activity [27,29]. It is well-established that the impairment of glial glutamate transporter function enhances the activation of AMPA and NMDA receptors in spinal dorsal horn neurons, as evidenced by the increased amplitudes and durations of AMPA and NMDA currents upon the pharmacological blocking of glial glutamate transporters [105,106,107]. Taken together, these patch clamp findings clearly highlight the heightened glutamatergic synaptic activity in spinal dorsal horn neurons (i.e., central sensitization) as a significant contributor to the genesis of chronic pain associated with SLE. Thus, targeting dysfunctional glutamatergic synapses in the spinal dorsal horn represents a viable approach to mitigate the excessive neuronal activation in this key relay center for pain signaling transmission from the periphery.

## 6. Signaling Molecules Regulating Spinal Glutamatergic Synaptic Activity and Chronic Pain in Lupus Mice

Recent studies conducted by our team and other researchers have uncovered a correlation between heightened spinal glutamatergic synaptic activity in female lupus mice experiencing chronic pain and the activation of microglia and astrocytes within the spinal dorsal horn [26,27]. Simultaneously, alterations in the production of proteins such as IL-1β, interleukin 18 (IL-18), cathepsin-B, p38 mitogen-activated protein kinase (p38 MAPK), colony-stimulating factor-1 (CSF-1), thrombin, protease-activated receptor 1 (PAR1), and AMPK activity have been observed in this region. Investigations have been undertaken to elucidate the functional implications of these altered signaling molecules on pain perception and glutamatergic synaptic transmission in the spinal dorsal horn of lupus mice with chronic pain (Table 2).

### 6.1. IL-1β and IL-18

It has been reported that the activation of glial cells in the spinal dorsal horn of lupus mice experiencing chronic pain was accompanied by the increased production of IL-1β [26,27], IL-18, and cathepsin B [26]. Both IL-1β and IL-18 belong to the IL-1 family [103], while cathepsin B is a protease implicated in the conversion of pro-IL-1β and IL-18 into their mature forms [111,112]. The involvement of IL-1β and IL-18 in the development of chronic pain induced by SLE was evident in experiments where spinal topical applications of IL-1β [27] or IL-18 [26] antagonists prolonged the latency of hind paw withdrawal responses to radiant heat stimuli in lupus mice. Additionally, intrathecal injection of recombinant IL-1β [113,114] or IL-18 [115,116] enhanced hind paw sensitivity to mechanical and heat stimulation. The overproduction of IL-1β has been observed in the spinal dorsal horn of animals with other pathological pain conditions, including neuropathic pain [89,117], inflammatory pain [118,119], arthritis [120,121], bone cancer pain [116,122], and chemotherapy (like paclitaxel)-induced chronic pain [123,124]. IL-1β is synthesized and released from microglia [124,125,126] and astrocytes [126,127]. IL-1β receptors are expressed in neurons (including nociceptive sensory afferents [128] and dorsal horn neurons [129]), astrocytes [130], and activated spinal microglia [131]. IL-1β receptor protein expression is enhanced under pathological pain conditions [130,132]. Patch clamping whole-cell recordings from spinal slices showed that the exogenous application of IL-1β into the spinal slice resulted in increased frequencies of miniature AMPA currents, indicating that IL-1β promotes glutamate release from the nociceptive primary afferents in the spinal dorsal horn [133]. Further experimental data demonstrated that IL-1β promotes this presynaptic glutamate release by enhancing presynaptic NMDA receptor activity [117,133]. IL-1β enhances the function of AMPA and NMDA receptors at the postsynaptic neurons at the first synaptic junction between nociceptive primary afferents and neurons in the spinal superficial dorsal horn [117,134,135]. Additionally, long-term potentiation in the spinal dorsal horn induced by high-frequency stimulation of the spinal dorsal root is significantly enhanced by exogenous IL-1β [125]. In lupus mice with chronic pain, IL-1β in the spinal dorsal horn enhances glutamatergic synaptic activity by suppressing glial glutamate transporter activity [27]. It has been demonstrated that IL-1β produces such effects by activating protein kinase C, which promotes the endocytosis of glial glutamate transporters in astrocytes [108]. Furthermore, the suppression of glial glutamate transporter activity in the spinal dorsal horn also attenuates GABAergic inhibitory synaptic activity by reducing presynaptic GABA synthesis through the glutamate–glutamine cycle and reducing GABAA receptor activity in postsynaptic neurons [136]. Therefore, the overproduction of IL-1β in the spinal dorsal horn enhances the activity in spinal neurons along the pain signaling pathway by increasing glutamatergic synaptic activity and attenuating inhibitory GABAergic synaptic activity, leading to enhanced pain perception.

The heightened production of IL-18 in the spinal dorsal horn not only contributes to the genesis of chronic pain induced by lupus but also in cases resulting from nerve injury [115,137] and bone cancer [116,138,139]. The overproduction of IL-18 in spinal microglia is associated with the genesis of long-term potentiation in the spinal cord induced by tetanic stimulation of the sciatic nerve [140]. IL-18 is primarily produced and released from microglia [138,141], while its receptors are located in astrocytes and neurons, including nociceptive primary afferents [26,138,142]. Patch-clamp whole-cell recordings have shown that the bath perfusion of IL-18 antagonists (IL-18BP) attenuated the frequency but not the amplitude of miniature AMPA currents recorded from spinal slices obtained from lupus mice with chronic pain [26]. In contrast, the bath perfusion of IL-18BP did not alter either the frequency or amplitude of miniature AMPA currents recorded from spinal slices from normal control mice [26]. These findings indicate that the enhanced IL-18 activity in lupus mice within the spinal dorsal horn enhances glutamate release from presynaptic nociceptive afferents, ultimately contributing to the thermal hyperalgesia observed in lupus mice [26].

### 6.2. p38 MAPK

p38 MAPK is a pivotal kinase involved in the activation of microglia, the production of pro-nociceptive mediators in the spinal cord, the genesis of pathological pain induced by nerve injury [138,143], inflammation [144], and bone cancer [138,143]. Heightened protein expression of phosphorylated p38 MAPK (increased p38 MAPK activity) in the spinal dorsal horn was observed in lupus mice experiencing chronic pain, coinciding with microglial activation in the same area. Importantly, the involvement of p38 MAPK in the genesis of chronic pain in lupus mice was confirmed in behavioral experiments, where the intrathecal injection of a p38 MAPK inhibitor (SB203580) [145] in lupus mice mitigated thermal hyperalgesia [26].

### 6.3. CSF-1

CSF-1 is implicated in the suppression of glial glutamate transporter activity in the spinal dorsal horn and thermal hyperalgesia in lupus mice [27]. Operating as a cytokine, CSF-1 exerts its influence by binding to the CSF-1 receptor (CSF-1R), a type III receptor tyrosine kinase [146]. Notably, CSF-1 plays a pivotal role in regulating the survival, proliferation, and differentiation of mononuclear cells, macrophages, and microglia [147]. Previous reports suggested that CSF-1 is produced from primary sensory neurons [148,149], and its receptors (CSF-1R) in the spinal dorsal horn are located in the microglia [150,151]. In lupus mice experiencing chronic pain, there was an increased protein expression of CSF-1 in the spinal dorsal horn [27]. The role of CSF-1 in the genesis of thermal hypersensitivity in lupus mice was evaluated by measuring the latencies of hind paw withdrawal responses to radiant heat stimuli before and after the intrathecal injection of a CSF-1R inhibitor (GW2580). The intrathecal injection of GW2580 attenuated thermal hyperalgesia, as evidenced by the prolonged latencies of hind paw withdrawal responses to radiant heat stimulation at 30 min after injection [27]. Furthermore, native control mice receiving an intrathecal injection of recombinant CSF-1 developed thermal hyperalgesia within 30 min of injection [27]. These findings align with previous studies showing that peripheral nerve injury induced the upregulation of CSF-1 production in the cell bodies of primary afferent neurons, and the intrathecal injection of recombinant CSF-1 stimulated microglial proliferation and induced mechanical allodynia in rats and mice [150,151].

Molecular mechanisms underlying the role of CSF-1 in regulating the genesis of chronic pain in lupus mice were further elucidated by experiments demonstrating that CSF-1 regulates glutamatergic synaptic activity by altering glial glutamate transporter activities. Glial glutamate transporter activities recorded from astrocytes in the spinal dorsal horn obtained from lupus mice were enhanced upon bath-perfusion of the CSF-1R inhibitor (GW2580). Conversely, in spinal slices from normal control mice, bath-perfusion of recombinant CSF-1 significantly reduced glial glutamate transporter activities [27]. Given that CSF-1 receptors are expressed exclusively in microglia in the spinal dorsal horn [150,151], and microglial activation leads to the release of inflammatory cytokines such as IL-1β [124,125,126], we tested whether IL-1β mediates the effects of CSF-1 on glial glutamate transporters. Glutamate transporter currents recorded from spinal slices of normal control mice demonstrated that recombinant CSF-1 significantly suppressed spinal glial transporter activity in astrocytes. This suppression was reversed by the addition of the IL-1 receptor blocker IL-1ra in the presence of CSF-1. Conversely, in spinal slices from lupus mice, following confirmation of the enhanced glial glutamate transporter activity by the CSF-1R inhibitor, the additional inhibition of IL-1β through bath perfusion of IL-1ra did not produce a significant alteration in glial transporter activity [27]. This study suggests that enhanced CSF-1 signaling activity in the spinal dorsal horn leads to thermal hypersensitivity in lupus mice by releasing IL-1β from microglia and suppressing glial glutamate transporter activity [27]. Furthermore, it is conceivable that CSF-1 also enhances spinal presynaptic glutamate release and postsynaptic glutamate receptor activity by releasing IL-1β, as described in Section 6.1.

### 6.4. Thrombin, PAR1, and AMPK

Recent studies indicated that thrombin plays a regulatory role in the heightened glutamatergic synaptic activity in the spinal dorsal horn, as well as in mechanical allodynia and thermal hyperalgesia in lupus mice [29]. Thrombin, an enzyme responsible for converting fibrinogen to fibrin in the blood coagulation pathway, exhibits diverse effects beyond its traditional role. Within the central nervous system (CNS), thrombin can act as an upstream inducer of proinflammatory cytokine production in the genesis of neuroinflammation observed in several pathological conditions, including neurodegenerative diseases, traumatic brain injury [152], and pathological pain conditions induced by nerve injury [153,154] and inflammation [155]. The thrombin receptor, protease-activated receptor 1 (PAR1), is located in neurons, astrocytes, and microglia within the CNS [156,157,158,159]. It was reported that the protein expression of thrombin and PAR1 in the spinal dorsal horn was increased in lupus mice with chronic pain. The intrathecal injection of a selective PAR1 blocker (SCH79797) attenuated mechanical and thermal hypersensitivity in lupus mice, while the intrathecal injection of thrombin induced mechanical and thermal hypersensitivity in normal control mice [29].

The impacts of the thrombin-PAR1 signaling pathway on spinal glutamatergic synaptic transmission were investigated by whole-cell patch recordings from spinal slices [29]. Bath-perfusion of the PAR1 blocker (SCH79797) reduced both the frequency and amplitude of miniature AMPA currents recorded from superficial dorsal horn neurons in spinal slices taken from lupus mice with chronic pain. In contrast, the bath application of thrombin enhanced the frequency and amplitude of miniature AMPA currents recorded from superficial dorsal horn neurons in spinal slices from normal control mice [29]. Thrombin bath perfusion resulted in a reduction in glial glutamate transporter activity recorded from spinal slices of normal control mice, while the reduction in glial glutamate transporter activity in lupus mice was enhanced upon the blockade of PAR1 with bath perfusion of SCH79797 [29]. These data indicate that thrombin potentiates spinal glutamatergic synaptic activity by enhancing presynaptic glutamate release, enhancing AMPA receptor function in the postsynaptic neurons, and suppressing glutamate clearance via suppressing glial glutamate transporter activity. Taken together, these studies suggest that the enhanced thrombin-PAR1 signaling activity in the spinal dorsal horn leads to the increased activation of glutamatergic synaptic activity and chronic pain in lupus mice [29].

Moreover, the protein expression of phosphorylated AMPK, indicative of AMPK activity, was found to decrease in the spinal dorsal horn of lupus mice experiencing chronic pain [29]. A similar suppression of AMPK activity was observed within the spinal dorsal horn of animals with nerve injury [89,160], bone cancer pain [161], and arthritis [120]. Within the CNS, AMPK is expressed in neurons, astrocytes, and microglia [162,163,164]. It has been demonstrated that the reduced AMPK activity in the spinal dorsal horn regulates the development of mechanical allodynia and thermal hyperalgesia by enhancing spinal glutamatergic synaptic activity and reducing glial glutamate transporter activity in lupus mice with chronic pain [29]. Interestingly, the suppressive effects induced by the exogenous application of thrombin [29] and IL-1β [89] on the spinal glial glutamate transporter activity in spinal slices were reversed via the AMPK activator (AICAR). AICAR prevented the endocytosis of spinal glial glutamate transporters induced by IL-1β [89]. These findings suggest that AMPK is a downstream signaling molecule used by thrombin and IL-1β to modulate glutamatergic synaptic activity in the spinal dorsal horn and chronic pain in animals with lupus disease.

### 6.5. GPR109A

We recently identified and characterized the role of a Gi-protein-coupled receptor (GPCR), GPR109A, in the regulation of spinal inflammation and chronic pain in lupus mice [26]. Originally discovered in adipocytes, GPR109A’s activation by niacin suppresses the release of free fatty acids and alleviates dyslipidemia [165,166,167]. Recent studies have recognized GPR109A as a crucial receptor that modulates inflammatory responses across various immune cell types, including microglia and macrophages [168,169,170,171]. The activation of GPR109A in both macrophages and microglia cell cultures has been shown to decrease the production of chemokines and pro-inflammatory cytokines induced by toll-like receptor 4 activation [171,172,173,174]. We found that GPR109A was expressed in microglia but not in astrocytes or neurons within the spinal dorsal horn [26]. Significantly, its expression showed a notable increase in lupus mice experiencing thermal hyperalgesia [26]. The intrathecal injection of a selective GPR109A agonist (MK1903) significantly attenuated thermal hyperalgesia in lupus mice, while this treatment did not alter the sensory perception in normal control mice. The activation of spinal GPR109A with MK1903 resulted in the attenuation of p38 MAPK activity and glutamatergic synaptic activity by suppressing the production of IL-18 and IL-1β in the spinal dorsal horn of lupus mice with chronic pain. These findings provide compelling evidence supporting the potential of targeting microglial GPR109A as a potent strategy for reversing spinal neuroinflammation, correcting abnormal excitatory synaptic activity, and managing chronic pain induced by SLE [26].

## 7. Conclusion Remarks and Prospectives

Managing chronic pain in patients with SLE remains a clinical challenge due to the limited effectiveness and safety profiles of currently available analgesics. Identifying signaling molecules that regulate the aberrant activation of neurons along the pain signaling pathway is a crucial step toward developing novel analgesics. Recent studies by our research group and others have uncovered key mechanisms underlying chronic pain in lupus mice. It has been reported that primary nociceptive neurons in lupus mice exhibit elevated resting membrane potentials and reductions in action potential thresholds and rheobases, leading to hyperexcitability. These changes are associated with the increased protein expression of proinflammatory cytokines such as TNFα and IL-1β. Further investigations have demonstrated that reduced AMPK activity in primary nociceptive neurons contributes to this increased membrane excitability in lupus mice with chronic pain. In the spinal dorsal horn, glutamatergic synaptic activity between primary nociceptive afferent terminals and spinal dorsal horn neurons is augmented by enhanced presynaptic glutamate release, postsynaptic glutamate receptor function, and reduced glial glutamate transporter activity. This enhanced glutamatergic synaptic activity and the resulting chronic pain in lupus mice are attributed to the overproduction of proinflammatory cytokines (such as IL-1β and IL-18), CSF-1, and thrombin, as well as reduced AMPK activity in the same region. Activation of the microglial GPR109A receptor in the spinal dorsal horn attenuates glutamatergic synaptic activity and hypersensitivity in lupus mice by suppressing the overproduction of spinal IL-1β and IL-18. Taken together, these findings highlight that targeting signaling molecules such as AMPK, IL-1β, IL-18, CSF-1, thrombin, and GPR109A holds promise for effectively reversing peripheral or central sensitization along the pain signaling pathway, ultimately offering relief from chronic pain induced by SLE. Figure 2 depicts a schematic illustration of the primary signaling molecular and synaptic mechanisms governing the excitability of DRG nociceptive neurons and the glutamatergic synaptic activity in the spinal dorsal horn of lupus mice experiencing chronic pain.

Our understanding of the cellular, synaptic, and molecular mechanisms underlying the genesis of pathological pain primarily stems from various animal models designed to mimic different clinical conditions. Numerous studies have investigated animal models of neuropathic pain induced by nerve injury or chemotherapy [175,176,177], inflammatory pain induced by CFA or carrageenan [177,178], bone cancer pain [177,178,179], arthritic pain [177,180], postoperative pain [177,181], and visceral pain [177,182]. However, our comprehension of the signaling molecules governing the pathology induced by SLE in the context of chronic pain is still in its nascent stage. Only four animal studies have been published on chronic pain induced by SLE, indicating a significant gap in this research field. Given that the etiology of SLE differs fundamentally from other disease entities, the signaling molecular mechanisms underlying chronic pain found in other animal models cannot be directly applied to chronic pain induced by SLE. Numerous avenues remain to be explored in this field, including investigations into sensory transduction, ion channels in peripheral sensory neurons, spinal and supraspinal synaptic plasticity, descending control, and their contribution to chronic pain induced by SLE. As outlined in this review, chronic pain induced by SLE appears to involve distinct mechanisms compared to chronic pain caused by other diseases. For instance, nociceptive sensory neurons in lupus mice exhibit some unique features not observed in other chronic pain animal models (see Table 1 and Section 3.3 for details). Furthermore, our studies have demonstrated that thermal hyperalgesia and impaired glial glutamate transporter function in female lupus-prone mice can be alleviated by blocking CSF-1 receptors on microglia (see Section 6.3 for details). The activation of microglial GPR109A receptors in the spinal cord mitigates thermal hyperalgesia in female lupus mice by suppressing the production of IL-1β and IL-18, as well as p38 MAPK (see Section 6.5 for details). Previous studies have indicated that microglial activation significantly contributes to the development of mechanical allodynia induced by nerve injury [183], the subcutaneous injection of formalin [184], or collagen-induced arthritis [185] in male animals, but not in females. Additional research is warranted to explore potential sexual dimorphism in the mechanisms underlying chronic pain induced by SLE. Due to our current limited understanding of the mechanisms driving the development of chronic pain in SLE, extensive efforts are required to explore novel signaling molecules that govern abnormal neuronal activity along the pain signaling pathway in both sexes. Ultimately, these endeavors could lead to the discovery and development of innovative analgesics tailored to effectively manage chronic pain in SLE patients.

## Figures and Tables

**Figure 1 ijms-25-03602-f001:**
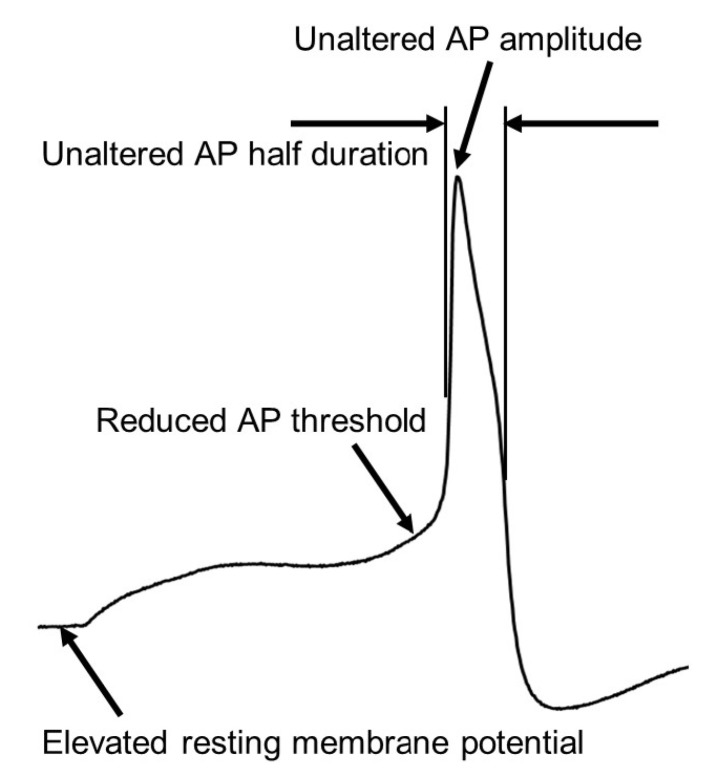
Schematic illustration for altered membrane properties in nociceptive primary sensory neurons in the DRG in lupus mice with chronic pain. Lupus mice with chronic pain exhibit elevated resting membrane potentials, a reduced action potential (AP) activating thresholds and rheobases, and lowered capacitance, while displaying no changes in AP half-duration and amplitude compared to normal control mice. These changes result in hyperexcitability in nociceptive primary sensory neurons in lupus mice [28].

**Figure 2 ijms-25-03602-f002:**
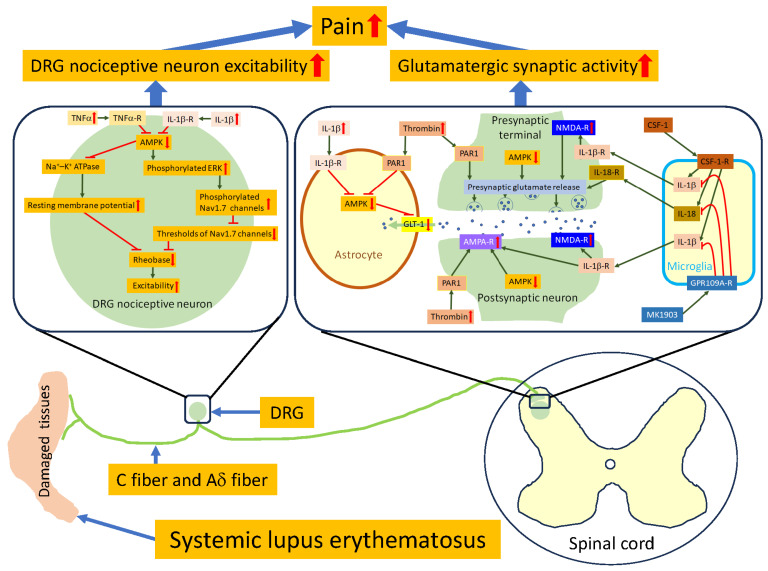
A schematic illustration of the primary signaling molecular and synaptic mechanisms governing the excitability of DRG nociceptive sensory neurons and the glutamatergic synaptic activity in the spinal dorsal horn of mice with chronic pain induced by SLE. In lupus mice with chronic pain, DRG nociceptive neurons exhibit increased excitability. This is attributed to the elevated production of TNFα and IL-1β in the DRG, which suppress AMPK activity. Consequently, AMPK suppression leads to inhibition of the Na^+^-K^+^ ATPase pump in the plasma membrane, raising the resting membrane potential. Additionally, reduced AMPK activity activates ERK, phosphorylating Nav. 1.7 channels and lowering their activation threshold. The combination of elevated resting membrane potential and reduced Nav. 1.7 activation threshold decreases neuronal rheobase, resulting in hyperexcitability of DRG nociceptive sensory neurons (i.e., peripheral sensitization). Furthermore, in the spinal dorsal horn of lupus mice with chronic pain, there is increased glutamatergic synaptic activity. The elevated expression of IL-1β, IL-18, CSF-1, and thrombin, along with reduced AMPK activity, contributes to this phenomenon. The increased activation of CSF-1 receptors (CSF-1-R) on microglia promotes IL-1β and IL-18 production. Presynaptic glutamate release is augmented by IL-1β, IL-18, CSF-1, thrombin, and reduced AMPK activity. IL-1β enhances presynaptic glutamate release by activating presynaptic NMDA receptors (NMDA-R). IL-1β also potentiates AMPA receptor (AMPA-R) and NMDA receptor (NMDA -R) activity in postsynaptic neurons. Moreover, the activation of IL-1β receptors (IL-1β-R) and thrombin receptor (PAR1) on astrocytes suppresses glial glutamate transporter (GLT-1) function by inhibiting AMPK activity. The activation of PAR1 or reduced AMPK activity at postsynaptic neurons leads to enhanced AMPA receptor activity in postsynaptic neurons. Overall, the increased glutamatergic activity in the spinal dorsal horn indicates central sensitization in lupus mice with chronic pain. Finally, the activation of GPR109A by the selective GPR109A agonist (MK1903) on microglia suppresses the enhanced glutamatergic activity by suppressing the production of IL-1β and IL-18. Symbols: ↑ increased; ↓ decreased.

**Table 1 ijms-25-03602-t001:** Membrane properties in nociceptive primary sensory neurons in the DRG of different pain animal models in comparison to normal controls.

Membrane Properties	SLE [28]	Neuropathic [56,57,65,66]	Arthritis [60,64]	Bone Cancer [61,62]	CFA [63,67]
Resting potential	↑ *	↑	↑	↑	↔
Capacitance	↓	?	↔	?	↔
Resistance	↔	?	?	↓	?
AP threshold	↓	↓	↓	↓	↓
Rheobase	↓	↓	?	↓	↓
AP duration	↔	↑	↑	↓	↓
AP amplitude	↔	↑	↔	↓	↑
AP spontaneous activity	↔	↑	↑	↓	↑

* ↑: increased; ↓: decreased; ↔: unchanged; ?: unknown; AP: action potential.

**Table 2 ijms-25-03602-t002:** Altered signaling molecules and their impacts on glutamatergic synaptic activity in the spinal dorsal horn of lupus mice with chronic pain.

Protein Expression of Signaling Molecules in the Spinal Dorsal Horn	Impacts on Glutamatergic Synaptic Activity		
	Presynaptic Glutamate Release	Postsynaptic Glutamate Receptor Activity	Glial Glutamate Transporter Activity
IL-1b ↑ *	↑	↑	↓
IL-18 ↑	↑	↔	?
CSF-1 ↑	↑	↑	↓
Thrombin ↑	↑	↑	↓
Phosphorylated AMPK ↑	↑	↑	↓

* ↑: increased; ↓: decreased; ↔: unchanged; ?: unknown.

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
