# Peer review of "Emerging Molecular and Synaptic Targets for the Management of Chronic Pain Caused by Systemic Lupus Erythematosus"

_ijms, 2024, doi:10.3390/ijms25073602_

Round 1
Reviewer 1 Report
Comments and Suggestions for Authors
The manuscript ‘Emerging Molecular and Synaptic Targets for the Management of Chronic Pain Caused by Systemic Lupus Erythematosus’ by Han-Rong Weng, focused on recents studies on chronic pain in systemic lupus erythematosus (SLE) reveal heightened neuronal excitability and increased glutamatergic activity in spinal neurons. Dysregulated molecules such as TNFα, IL-1β, ERK, AMPK, IL-18, CSF-1, and thrombin contribute to these alterations. Targeting spinal GPR109A receptors in microglia shows promise in alleviating pain by modulating inflammation and reducing glutamatergic activity. These findings offer potential avenues for developing analgesics for SLE-induced chronic pain. Still some issues need to be clarified, as listed below, before the manuscript can be accepted for publication in Int.J.Mol.Sci.
1. Please provide a more comprehensive discussion in Section 2 by including information on alternative animal models other than mice.
2. To adequately illustrate the hyperexcitability of nociceptive primary sensory neurons in the dorsal root ganglion from lupus mice with chronic pain, please include supplementary figure in section 3.
3. Certainly, to break down the lengthy Section 3 and highlight its importance, it’s advisable to add subheadings summarizing each paragraph.
4. Please summarize the regulatory signaling molecules in tabular form.
Author Response
Reviewer 1:
The manuscript ‘Emerging Molecular and Synaptic Targets for the Management of Chronic Pain Caused by Systemic Lupus Erythematosus’ by Han-Rong Weng, focused on recents studies on chronic pain in systemic lupus erythematosus (SLE) reveal heightened neuronal excitability and increased glutamatergic activity in spinal neurons. Dysregulated molecules such as TNFα, IL-1β, ERK, AMPK, IL-18, CSF-1, and thrombin contribute to these alterations. Targeting spinal GPR109A receptors in microglia shows promise in alleviating pain by modulating inflammation and reducing glutamatergic activity. These findings offer potential avenues for developing analgesics for SLE-induced chronic pain. Still some issues need to be clarified, as listed below, before the manuscript can be accepted for publication in Int.J.Mol.Sci.
Please provide a more comprehensive discussion in Section 2 by including information on alternative animal models other than mice.
Responses:
It should be noted that murine models remain pivotal in experimental research on SLE. The use of other animal species for SLE study is rare [1]. I underscored this fact by adding:
On page 4, line 10, I added that “Murine models serve as the cornerstone of experimental animals utilized to identify potential therapeutic targets within the various signaling pathways dysregulated in SLE. A range of mouse models for SLE, encompassing those that spontaneously develop the condition, as well as those induced or genetically engineered, have been extensively employed in research endeavors [2, 3]”.
- To adequately illustrate the hyperexcitability of nociceptive primary sensory neurons in the dorsal root ganglion from lupus mice with chronic pain, please include supplementary figure in section 3.
Responses:
I have added a new figure (Figure 1) with legend (see below) in section 3 to illustrate changes of membrane properties in nociceptive primary sensory neurons in the dorsal root ganglion from lupus mice with chronic pain
Figure 1. Schematic illustration for altered membrane properties in nociceptive primary sensory neurons in the DRG in lupus mice with chronic pain. Lupus mice with chronic pain exhibit elevated resting membrane potentials, reduced action potential (AP) activating thresholds and rheobases (not shown), and lowered capacitance (not shown), while displaying no changes in AP half duration and amplitude compared to normal control mice.
- Certainly, to break down the lengthy Section 3 and highlight its importance, it’s advisable to add subheadings summarizing each paragraph.
Responses:
Three subheadings (see below) have been added to summarizing each paragraph and highlight the main points.
3.1. Resting membrane potential is elevated while capacitance is reduced in nociceptive primary sensory neurons of lupus mice with chronic pain
3.2. Rheobase and action potential threshold are reduced in nociceptive primary sensory neurons of lupus mice with chronic pain
3.3. Comparison of membrane properties in nociceptive primary sensory neurons in animals with lupus-induced pain and pain induced by other diseases (Table 1)
- Please summarize the regulatory signaling molecules in tabular form.
Responses:
Table 2 has been added.
References
- Chen J, Liao S, Zhou H, Yang L, Guo F, Chen S, et al. Humanized Mouse Models of Systemic Lupus Erythematosus: Opportunities and Challenges. Front Immunol. 2021;12:816956. Epub 2022/02/05. doi: 10.3389/fimmu.2021.816956. PubMed PMID: 35116040; PubMed Central PMCID: PMCPMC8804209.
- Halkom A, Wu H, Lu Q. Contribution of mouse models in our understanding of lupus. Int Rev Immunol. 2020;39(4):174-87. Epub 2020/03/24. doi: 10.1080/08830185.2020.1742712. PubMed PMID: 32202964.
- Li W, Titov AA, Morel L. An update on lupus animal models. Curr Opin Rheumatol. 2017;29(5):434-41. Epub 2017/05/26. doi: 10.1097/BOR.0000000000000412. PubMed PMID: 28537986; PubMed Central PMCID: PMCPMC5815391.

Reviewer 2 Report
Comments and Suggestions for Authors
This is an excellent review that provides a detailed description of the mechanisms and factors involved in chronic pain caused by systemic lupus erythematosus (SLE). The sections in which the review is divided are appropriate as they allow for an understanding of the models used in SLE and subsequently the key molecules and mechanisms associated with chronic pain in SLE models. However, there are some points to be considered in this manuscript:
1. Lines 69-76 are redundant, as they only list the section names that are described later. Instead, the author can globally mention what they intend to convey in those sections.
2. Abbreviations are described multiple times. Remember that once a word is described for the first time, the abbreviation should be used thereafter. For example, DRG is described in line 111 but is repeated in lines 132 and 135, as well as GTCs.
3. Despite being a well-structured review, I believe that in section 4, a table summarizing the main targets and their mechanisms could be added.
Author Response
Reviewer 2
This is an excellent review that provides a detailed description of the mechanisms and factors involved in chronic pain caused by systemic lupus erythematosus (SLE). The sections in which the review is divided are appropriate as they allow for an understanding of the models used in SLE and subsequently the key molecules and mechanisms associated with chronic pain in SLE models. However, there are some points to be considered in this manuscript:
- Lines 69-76 are redundant, as they only list the section names that are described later. Instead, the author can globally mention what they intend to convey in those sections.
Responses:
The original lines 69-76 have been removed, and replaced with the following sentences “In this review, I have summarized recent advancements in our comprehension of the mechanisms driving peripheral and spinal central sensitization in a mouse model of SLE with chronic pain. It's anticipated that these studies lay the groundwork and offer pathways for the development of innovative analgesics to manage chronic pain induced by SLE.”
- Abbreviations are described multiple times. Remember that once a word is described for the first time, the abbreviation should be used thereafter. For example, DRG is described in line 111 but is repeated in lines 132 and 135, as well as GTCs.
Responses: Corrected.
- Despite being a well-structured review, I believe that in section 4, a table summarizing the main targets and their mechanisms could be added.
Responses:
I have added Figure 1 and its legend (see below) and Table 1 in section 3, and Table 2 in section 6, and the main targets and their mechanisms in section 4 have been summarized in Figure 2, which is referred in the text of section 4.

Reviewer 3 Report
Comments and Suggestions for Authors
The aim of the evaluated paper is to provide a review about molecular and synaptic targets for the management of chronic pain caused by Systemic Lupus Erythematosus (SLE).The subject is not new, but is one of great interest in SLE management. The article is interesting, well documented and it cites a spectacular number of papers (177) but it has to be reorganised in order to be easier to follow. It has just one image placed on the final part but I suggest to add several more at previous chapter.
Keywords – please update them with systemic lupus and chronic pain
Introductory part provides some data about lupus that has to be updated:
- Row 43 – etiopathogenesis of SLE is not only about autoantibodies
- Row 48 – since we are speaking about epidemiology please do not resume only at North America
- Rows 56-61 please mention that neither North American nor European guidelines have recommendations related to the pain management. Also there are no randomised clinical trials addressing this issue
- Rows 68-76 are probably unnecessary
Please add one table for each chapter 2 and 3 in order to summarize existing studies and their results
Described studies are specifying the sex of the mice - what was the reason for choosing the sex of the experimental animal?
Author Response
Reviewer 3:
The aim of the evaluated paper is to provide a review about molecular and synaptic targets for the management of chronic pain caused by Systemic Lupus Erythematosus (SLE). The subject is not new, but is one of great interest in SLE management. The article is interesting, well documented and it cites a spectacular number of papers (177) but it has to be reorganised in order to be easier to follow. It has just one image placed on the final part but I suggest to add several more at previous chapter.
Responses:
I have added Figure 1 and Table 1 in section 3, and Table 2 in section 6.
Keywords – please update them with systemic lupus and chronic pain
Responses:
Both “systemic lupus” and “chronic pain” have been mentioned in the title and abstract, making them easily discoverable through internet searches. I lean towards excluding these terms in the key word list to prevent redundancy.
Introductory part provides some data about lupus that has to be updated:
- Row 43 – etiopathogenesis of SLE is not only about autoantibodies
Responses:
On page 3, line 2, the original sentence “Systemic lupus erythematosus (SLE) is an autoimmune disease characterized by production of autoantibodies that attack multiple organs and tissues” has been removed. This is replaced with updated concept about the etiopathogenesis of SLE with three recent references as follow: “Systemic lupus erythematosus (SLE) is a chronic autoimmune disease marked by dysregulation of adaptive and innate immunity. Consequently, loss of self-tolerance and formation of nuclear autoantigens and immune complexes results in inflammation and damage of multiple organs [1-3].
Row 48 – since we are speaking about epidemiology please do not resume only at North America
Responses:
On page 3, line 6, I have replaced “It is estimated that there are 2.4 SLE patients per 1000 people In North America [4, 5].”with “It is estimated that there are 61.08 adults with SLE per 100,000 persons, corresponding to approximately 3.17 million adults worldwide [6].”
- Rows 56-61 please mention that neither North American nor European guidelines have recommendations related to the pain management. Also there are no randomised clinical trials addressing this issue
Responses:
As a neurobiologist, I am not inclined to comment on clinical guidelines or trials for managing chronic pain in lupus patients. However, my review article emphasizes the clinical challenges in managing chronic pain in SLE, particularly due to the limited safety or efficacy profiles of the existing analgesics. Nonetheless, I have come across the following recommendations in Europe and guidelines in the USA online.
European League Against Rheumatism (EULAR)**: EULAR provides recommendations for the management of systemic lupus erythematosus (SLE), including strategies for pain management. (See https://ard.bmj.com/content/83/1/15)
American College of Rheumatology (ACR)**: ACR offers guidelines for the management of lupus, including pain management strategies (See: https://assets.contentstack.io/v3/assets/bltee37abb6b278ab2c/blt244ee5255b4b0499/lupus-guideline-project-plan-2025.pdf).
I would greatly appreciate it if Reviewer 1 could provide relevant references to support his statements, so that I can make changes in the text according to his/her comments.
Rows 68-76 are probably unnecessary
Responses:
The original lines 69-76 have been removed, and replaced with the following sentences “In this review, I have summarized recent advancements in our comprehension of the mechanisms driving peripheral and spinal central sensitization in a mouse model of SLE with chronic pain. It's anticipated that these studies lay the groundwork and offer pathways for the development of innovative analgesics to manage chronic pain induced by SLE.”
Please add one table for each chapter 2 and 3 in order to summarize existing studies and their results
Responses:
Figure 1, Table 1 and Table 2 and their legends have been added.
Described studies are specifying the sex of the mice - what was the reason for choosing the sex of the experimental animal?
Responses:
The reason for choosing the sex of the experimental animals has not been addressed in the current publications. In the “Closing remarks and prospectives” section, I have acknowledged such fact that “our comprehension of the signaling molecules governing the pathology induced by SLE in the context of chronic pain is still in its nascent stage. Only four animal studies have been published on chronic pain induced by SLE, indicating a significant gap in this research field.” Due to this significant gap, all the four published research did not address issues related to sexual dimorphism on SLE induced pain or the reason for choosing the sex of the experimental animals.
As a matter of fact, studies on animals of both sexes are needed. Therefore, in the “Closing remarks and prospectives” section (on page XX, line XX), I have called for the attention to address the issues related to sexual dimorphism by modifying the sentence that “extensive efforts are required to explore novel signaling molecules that govern abnormal neuronal activity along the pain signaling pathway” into “extensive efforts are required to explore novel signaling molecules that govern abnormal neuronal activity along the pain signaling pathway in both sexes.”
References
- Crow MK. Pathogenesis of systemic lupus erythematosus: risks, mechanisms and therapeutic targets. Ann Rheum Dis. 2023;82(8):999-1014. Epub 2023/02/16. doi: 10.1136/ard-2022-223741. PubMed PMID: 36792346.
- Fanouriakis A, Tziolos N, Bertsias G, Boumpas DT. Update omicronn the diagnosis and management of systemic lupus erythematosus. Ann Rheum Dis. 2021;80(1):14-25. Epub 2020/10/15. doi: 10.1136/annrheumdis-2020-218272. PubMed PMID: 33051219.
- Durcan L, O'Dwyer T, Petri M. Management strategies and future directions for systemic lupus erythematosus in adults. Lancet. 2019;393(10188):2332-43. Epub 2019/06/11. doi: 10.1016/S0140-6736(19)30237-5. PubMed PMID: 31180030.
- Rees F, Doherty M, Grainge MJ, Lanyon P, Zhang W. The worldwide incidence and prevalence of systemic lupus erythematosus: a systematic review of epidemiological studies. Rheumatology (Oxford). 2017;56(11):1945-61. Epub 2017/10/03. doi: 10.1093/rheumatology/kex260. PubMed PMID: 28968809.
- Pons-Estel GJ, Alarcon GS, Scofield L, Reinlib L, Cooper GS. Understanding the epidemiology and progression of systemic lupus erythematosus. Semin Arthritis Rheum. 2010;39(4):257-68. Epub 2009/01/13. doi: 10.1016/j.semarthrit.2008.10.007. PubMed PMID: 19136143; PubMed Central PMCID: PMCPMC2813992.
- Tian J, Zhang D, Yao X, Huang Y, Lu Q. Global epidemiology of systemic lupus erythematosus: a comprehensive systematic analysis and modelling study. Ann Rheum Dis. 2023;82(3):351-6. Epub 2022/10/15. doi: 10.1136/ard-2022-223035. PubMed PMID: 36241363; PubMed Central PMCID: PMCPMC9933169.
